# Evaluating Population Normalization Methods Using Chemical Data for Wastewater-Based Epidemiology: Insights from a Site-Specific Case Study

**DOI:** 10.3390/v17050672

**Published:** 2025-05-04

**Authors:** Marco Verani, Ileana Federigi, Alessandra Angori, Alessandra Pagani, Francesca Marvulli, Claudia Valentini, Nebiyu Tariku Atomsa, Beatrice Conte, Annalaura Carducci

**Affiliations:** 1Laboratory of Hygiene and Environmental Virology, Department of Biology, University of Pisa, Via S. Zeno 35/39, 56123 Pisa, Italy; marco.verani@unipi.it (M.V.); alessandra.angori@biologia.unipi.it (A.A.); alessandra.pagani@phd.unipi.it (A.P.); nebiyu.atomsa@phd.unipi.it (N.T.A.); beatrice.conte@biologia.unipi.it (B.C.); annalaura.carducci@unipi.it (A.C.); 2Acque S.p.A., Via A. Bellatalla 1, 56121 Pisa, Italy; f.marvulli@acque.net; 3Gaia S.p.A., Via G. Donizetti 16, 55045 Pietrasanta, Italy; claudia.valentini@gaia-spa.it

**Keywords:** environmental surveillance, clinical surveillance, chemical parameters, population marker, chemical oxygen demand, COD, biochemical oxygen demand, BOD_5_, ammonia, public health

## Abstract

Wastewater-based epidemiology (WBE) has been widely employed to track the spread of human pathogens; however, correlating wastewater data with clinical surveillance remains challenging due to population variability and environmental factors affecting wastewater composition. This study evaluated different SARS-CoV-2 normalization methods, comparing static population estimates with dynamic normalization based on common physicochemical parameters: chemical oxygen demand (COD), biochemical oxygen demand (BOD_5_), and ammonia (NH_4_-N). Wastewater samples were collected from four urban wastewater treatment plants (WWTPs) in northwestern Tuscany (Italy) from February 2021 to March 2023. The correlations between normalized viral loads and clinical COVID-19 cases were highest for static normalization (ρ = 0.405), followed closely by dynamic normalization using COD and BOD_5_ (ρ = 0.378 each). Normalization based on NH_4_-N was less effective. These findings suggest that chemical parameters, particularly COD and BOD_5_, offer a valid alternative for viral normalization when population estimates or flow rate measurements are unavailable. These parameters provide a cost-effective and practical approach for improving WBE reliability, particularly in resource-limited settings. Our results reinforce the importance of normalization in WBE to enhance its representativeness and applicability for public health surveillance.

## 1. Introduction

Since the spread of SARS-CoV-2 in 2020, a vast body of scientific research has been produced on wastewater-based epidemiology (WBE) to track its transmission. Variables influencing SARS-CoV-2 RNA concentrations in wastewater have been widely addressed, especially in terms of sampling and laboratory-related factors, i.e., sampling techniques, sample processing, and analytical methodology, which have been widely reviewed [1,2,3]. However, the viral signal in wastewater is intrinsically influenced by local catchment characteristics, e.g., sewage origin (i.e., relative contribution of industrial or urban discharges), sewer network type (combined, separated or hybrid configurations with storm drain connected to the sewage pipeline), and length (affecting the hydraulic retention time, thus influencing viral decay). Therefore, the total covered population size is of paramount importance to understand the human contribution to wastewater, thus allowing comparisons of viral levels across different locations (i.e., cities or nations) and time [4].

In the context of the WBE-related literature, the practice of the “population normalization” of SARS-CoV-2 concentrations (gene copies per volume of wastewater sample) allows to account for the number of individuals in the catchment area served by a certain wastewater treatment plant (WWTP), and to express the data as daily viral load (gene copies per person connected to the sewer system per day). Normalization based on population can follow either static or dynamic approaches. Static population estimates can be obtained using census data that represent the permanent residential population served by the sewer network flowing into a certain WWTP [5] or population equivalent (PE), which refers to the design capacity of the WWTP [6], although it may not accurately reflect the actual load entering the sewerage system. Thus, static population estimates assume a constant population size over time and do not account for transient fluctuations caused by births, deaths, and migrations, or those attributable to seasonal tourism or work–travel commuting [7].

On the other hand, dynamic population estimates address these changes by monitoring specific population markers in wastewater, defined as analytes that reflect anthropogenic contribution to wastewater [4]. Various population markers have been proposed for SARS-CoV-2 viral load estimation, which can be classified into different categories on the basis of their nature and origin:(i)Chemical parameters. can be associated with human excretions into wastewater, mainly feces that are inferred by parameters estimating organic matter content, e.g., chemical oxygen demand (COD), biological oxygen demand (BOD_5_), or urine that are reflected by nitrogen-containing compounds, e.g., total nitrogen, ammonia (NH_3_), and ammonium (NH_4_-N) [8,9,10,11];(ii)Endogenous chemical biomarkers derived from the metabolism of human molecules, e.g., creatinine [12,13], serotonin (5-hydroxy-indole acetic acid—5-HIAA [12,13,14]) or cholesterol (coprostanol [15]);(iii)Exogenous chemical biomarkers originating from the metabolic activity from food and beverage consumption, such as coffee (caffeine metabolites [12,13,16]) or artificial sweetener (acesulfame [17]);(iv)Biological biomarkers represented mainly by the human gut microbiophage CrAssphage [18,19] and the Pepper Mild Mottle Virus (PMMoV), a viral pathogen associated with peppers and their processed products [12,17,20].

Nowadays, there is no golden standard in WBE for the selection of population markers. Each of them has strengths and limitations, as reviewed by Boogaerts et al. (2024) [7]; thus, the choice mostly depends on the laboratory testing procedures and equipment.

Various institutions and public health authorities recommend using population-normalized viral loads rather than raw viral concentrations when comparing wastewater data with daily clinical case numbers (e.g., European Union [21], Public Health Agency of Canada [22]; United States Centers for Disease Control and Prevention [23]). However, correlations between WBE results and clinical surveillance data still remain challenging, owing to uncertainties that also affect clinical data [24,25,26]. In fact, clinical data collection can be influenced by testing methods (e.g., antigenic or molecular) and the magnitude of the testing rate, which can vary over time on the basis of local public health recommendations. Moreover, clinical surveillance generally shows a selection bias, since it focuses on symptomatic individuals, people with certain medical conditions that make them at risk for COVID-19, and in some cases, also focuses on people in contact with confirmed COVID-19 cases through contact tracing. Then, the intrinsic variability in case reports (e.g., reporting delay, imperfect geographic attribution of clinical cases count [27,28]) can distort the relationship with WBE data [29].

In this study, we assessed different SARS-CoV-2 normalization methods for wastewater samples using both static and dynamic approaches based on chemical parameters (COD, BOD_5_, NH_4_-N), with the following aims:(i)Comparing dynamic population normalization results with the standard normalization approach based on static population estimates;(ii)Evaluating correlations between WBE results obtained through different normalization methods and clinical COVID-19 case data in order to determine which SARS-CoV-2 normalization approach best aligns environmental and clinical data.

## 2. Materials and Methods

### 2.1. Time and Location of This Study

This study was carried out from February 2021 to March 2023 at four urban WWTPs located in northwestern Tuscany (Italy), where domestic wastewater production accounted for more than 98% of the total wastewater load [26]. These WWTPs were investigated as part of the surveillance network of the national project “Environmental Surveillance of SARS-CoV-2 by urban sewages in Italy” (SARI), as previously reported [26,30]. The sewerage system of each WWTP was completely characterized to define the WWTP catchment population and the percentage of different municipalities covered. In fact, the geographical coverage of the WWTPs did not always aligned with administrative boundaries and could span three to six municipalities (Table 1), including tourist areas in the cases of WWTP1 and WWTP4. The WWTPs slightly differed in terms of their sewer network structure. WWTP3 and WWTP4 were served by a separate sewerage system, meaning sewage and greywater were collected separately. Instead, in WWTP1 and WWTP2, the sewerage systems were a small percentage (2%) or for the most part (89%) combined [26]. The intrusion of rainwater runoff into sewage pipes, known as parasitic rainwater, has been described by the Integrated Water Service Managers and has been reported previously [31].

### 2.2. Wastewater Monitoring

Twenty-four hour composite wastewater samples were collected at the inlet of each WWTP and analyzed for SARS-CoV-2 by the Hygiene and Environmental Laboratories (Department of Biology, University of Pisa, Pisa) and for physicochemical parameters by the Integrated Water Service Managers. Sampling was performed on a weekly basis; however, on certain sampling dates, samples for viral analysis were not collected due to logistic reasons.

#### 2.2.1. SARS-CoV-2 Analysis

A total of 255 raw sewage samples were collected from the four WWTPs (n = 75, 76, 48, and 56 for WWTP1, WWTP2, WWTP3, and WWTP4, respectively) and analyzed for SARS-CoV-2 within the Italian national program (SARI project) for wastewater, using the analytical protocols described elsewhere [26,30,32]. Briefly, the procedure included the following phases: (i) inactivation of samples (50 mL) at 56 °C for 30 min for laboratory safety reasons; (ii) the addition of Mengovirus strain vMC0 (100 µL) as an external process control; (iii) solid removal by centrifugation at 4500× *g* for 30 min; (iv) viral particle precipitation by the addition of PEG8000 (4 g) and NaCl (0.9 g) to the supernatant (40 mL) followed by centrifugation at 12,000× *g* for 2 h; (v) viral particle recovery using 2 mL of lysis buffer containing guanidine thiocyanate; (vi) the extraction of viral RNA using NucliSense EasyMag (bioMérieux, Marcy l’Etoile, France), obtaining a final volume of 100 µL; (vii) purification of the extracted RNA using a commercial One-Step PCR Inhibitor Removal Kit (Zymo Research, Irvine, CA, USA); and (viii) detection of SARS-CoV-2 (target gene orf1b, nsp14) and of vMC0 genomes using real-time one-step qPCR reaction with the mixture AgPath-ID One-Step RT-PCR (Applied Biosystems—Thermo Fisher Scientific, Austin, TX, USA) and genome quantification using the standard curve obtained by the serial dilution (from 1.0 × 10^1^ GC/µL to 1.0 × 10^5^ GC/µL) of a synthetic dsDNA.

#### 2.2.2. Chemical Parameter Analysis

Chemical parameters were routinely monitored by the Integrated Water Service Managers and were therefore determined in the same wastewater samples analyzed for SARS-CoV-2, as well as in additional samples collected for the operational monitoring of the WWTPs. Overall, a total of 359 samples (n = 91, 93, 85, and 90 for WWTP1, WWTP2, WWTP3, and WWTP4, respectively) were collected and analyzed for COD, BOD_5_, and NH_4_-N. COD was measured according to a standard norm, which involved the addition of a reagent and subsequent spectrophotometric analysis (ISO 15705:2002 “Water quality—Determination of the chemical oxygen demand index (ST-COD)—Small-scale sealed-tube method”). BOD_5_ was obtained by inoculating nutrients and microorganisms into the sample and measuring the difference between the final and initial oxygen after 5 days of incubation (APAT CNR IRSA 5120 B2 Man 29 2003; UNI EN 1899-2:2000 “Water quality—Determination of biochemical oxygen demand after n days (BODn)—Method for undiluted samples”). Ammonia nitrogen was measured using a colorimetric test that quantified the concentration of ammonium ions in the water samples (HACH LANGE LCK 303).

### 2.3. SARS-CoV-2 Normalization Approaches

#### 2.3.1. SARS-CoV-2 Static Population Normalization

Viral normalization based on the static approach was conducted according to Equation (1)(1)$$static\;NVL=\frac{C_{RNA}\times F_{d}}{P}\times 1000$$
where static *NVL* is the normalized viral load (GC/day/1000 inhabitants) of each WWTP according to static population approach, *C_RNA_* is the concentration of SARS-CoV-2 obtained during wastewater monitoring (GC/L), *F_d_* is the daily wastewater flow rate of the WWTPs (L/day), *P* is the population served by each WWTP (number of inhabitants), and 1000 is a constant used to represent the viral load of 1000 inhabitants.

#### 2.3.2. SARS-CoV-2 Dynamic Population Normalization

Viral normalization based on the dynamic approach was performed following two normalization metrics according to Sweetapple et al. (2023) [10]. Briefly, the SARS-CoV-2 load was computed using a “low data approach” when only analytical wastewater data on the SARS-CoV-2 and chemical parameters were available and a “high data approach” when technical information on the sewer network and flow rate were also available, as detailed below.

##### 2.3.2.1. SARS-CoV-2 Dynamic Population Normalization Based on Low Data Approach

Viral load was calculated using Equation (2), which considered the concentrations of analytes in wastewater, namely SARS-CoV-2 and chemical parameters.(2)$$dynamic\;NVL_{i}^{ld}=\frac{C_{RNA}}{C_{c,i}}\times 1000$$
where $dynamic\;NVL_{i}^{ld}$ is the normalized viral load according to the low data approach (*ld*) for dynamic population normalization, where *i* indicates the type of chemical parameter (COD, BOD_5_, or NH_4_-N), *C_RNA_* is the concentration of SARS-CoV-2 obtained during wastewater monitoring (GC/L), *C_c,i_* is the concentration of each chemical parameter (mg/L), and 1000 is a constant used to represent the viral load of 1000 inhabitants. In the low data approach, the normalized viral load is expressed as the SARS-CoV-2 concentration per unit of a chemical parameter, which is assumed to be directly proportional to the SARS-CoV-2 gene copies per capita per day in urban WWTPs that receive predominantly domestic discharge rather than industrial [10].

##### 2.3.2.2. SARS-CoV-2 Dynamic Population Normalization Based on High Data Approach

Viral load was calculated using Equation (3), which considered the concentrations of analytes in wastewater as well as the WWTP influent flow rate and served population.(3)$$dynamic\;NVL_{i}^{hd}=\frac{C_{RNA}\times M_{c,i}}{C_{c,i}}\times 1000$$
where dynamic $dynamic\;NVL_{i}^{hd}$ is the normalized viral load (GC/day/1000 inhabitants) according to the high data approach (*hd*) for dynamic population normalization, where *i* indicates the type of chemical parameters (COD, BOD_5_, or NH_4_-N), *C_RNA_* is the concentration of SARS-CoV-2 obtained during wastewater monitoring (GC/L), *M_c,i_* is the mean amount of daily production per person (mg/day/inhabitant) for each chemical (*i*), *C_c,i_* is the concentration of each chemical parameter detected in wastewater (mg/L), and 1000 is a constant used to represent the viral load of 1000 inhabitants.

In this study, the parameter *M_c,i_* was calculated for each WWTP according to Equation (4) in order to obtain a site-specific daily marker discharge per capita (Table 2).(4)$$M_{c,i}=\frac{C_{c,i}\times F_{d}}{P}$$
where the concentration of each chemical parameter (*C_c,i_*, mg/L) and the daily influent flow rate (*F_d_*, L/day) were collected during the national lockdown period (February 2021–October 2021 [26]) when wastewater at the WWTP inlet was considered representative of the catchment area, given the suppression of population variability resulting from restrictive containment measures and the limited movement of people; and *P* is the population served by each WWTP (number of inhabitants; Table 1).

### 2.4. Meteorological Data

Data on precipitation were collected during the environmental monitoring period using the public database of the Regional Hydrological Service (SIR; https://www.sir.toscana.it/, accessed on 1 February 2025). Site-specific monitoring stations were selected for each WWTP based on their proximity to the plant (<5 km) and their ability to measure rainfall: TOS01000544 for WWTP1, TOS11000071 for WWTP2, TOS01004007 for WWTP3, and TOS02004091 for WWTP4.

For the environmental sampling data, rainfall was quantified as the total precipitation in the previous 24 h according to the definition of wet weather given by the Copernicus Climate Change Service (https://climate.copernicus.eu/ESOTC; accessed on 15 January 2025) [33].

### 2.5. Clinical Data

Clinical data were collected in a specific timeframe, referring to a period when clinical surveillance was particularly active in Tuscany (Italy), and only molecular tests were used to confirm positive COVID-19 cases (February 2021 to December 2021; [26]). Thus, in the context of this study, clinical cases referred to individuals who tested positive via molecular diagnostic tests, regardless of the presence of disease symptoms. The number of newly reported daily COVID-19 cases for each municipality was obtained from the Regional Health Agency of Tuscany (Agenzia Regionale della Sanita della Toscana, ARS) and they were used to calculate the number of new confirmed cases per week referred to each WWTP, according to Equation (5).(5)$$ACC_{WWTP}=\frac{\sum_{1}^{n}cases_{x}\times Fm_{x}}{7}$$
where *ACC_WWTP_* is the number of new positive cases within each WWTP’s catchment area in a week, *cases_x_* refers to the total number of COVID-19 clinical cases in each municipality (*x*) discharging into the WWTP during a week, *F_mx_* is the fraction of the population of that municipality served by the WWTP (%), *n* is the total number of municipalities served by the WWTP (Table 1), and 7 is a dividing factor used to obtain the average weekly *ACC_WWTP_* in order to make the clinical data comparable with normalized viral loads that are derived from weekly sampling [26].

### 2.6. Data Analysis

The distribution of chemical parameters for each WWTP was described using descriptive statistics, i.e., the median and interquartile range (IQR). Statistical analysis involved the Kruskal–Wallis test, Spearman’s correlation, and linear regression analysis. The Kruskal–Wallis test was applied to evaluate differences among WWTPs in terms of the influent wastewater flow rate and concentrations of chemical parameters, followed by Dunn’s multiple comparisons post hoc test.

Spearman’s correlation was performed to test the associations between flow rate, chemical data, and rainfall, tested separately for each WWTP and also by aggregating all data from the different WWTPs (hereafter referred to as pooled WWTPs). Spearman’s correlation was computed over logarithmically transformed values when testing the association between clinical COVID-19 cases and non-normalized SARS-CoV-2 values, as well as normalized SARS-CoV-2 loads using either the static method (*static NVL*) or dynamic methods based on low data ($NVL_{COD}^{ld}$, $NVL_{BOD5}^{ld}$, $NVL_{NH4+}^{ld}$) or high data ($NVL_{COD}^{hd}$, $NVL_{BOD5}^{hd}$, $NVL_{NH4+}^{hd}$) approaches.

Regression analysis was performed to investigate the relationship between rainfall (independent variable) and the WWTP influent flow rate (dependent variable), and then between the WWTP influent flow rate (independent variable) and each chemical parameter (dependent variables), performed separately for each WWTP and on pooled WWTPs. As part of the regression analysis, ordinary least squares (OLS) regression was performed for each WWTP and for the pooled dataset, and residual autocorrelation was tested using the Durbin–Watson (DW) test, whose values ranged from 0 (positive autocorrelation) to 4 (negative autocorrelation) [34]. In cases where significant autocorrelation was detected, generalized least squares (GLS) models were applied to provide more robust and reliable coefficient of determination (R^2^) estimates.

Statistical significance was traditionally considered at a *p*-value of 0.05. All statistical analyses and figures were performed and created using R software version 4.3.2.

## 3. Results

### 3.1. Description of Chemical Parameters

The results of the wastewater monitoring are reported in Appendix A and summarized in Table 3, separately for each WWTP. Wastewater flow showed differences among the WWTPs, depending on the size of the served population: the higher the population size, the higher the influent flow rate (WWTP3 > WWTP2 > WWTP4 > WWTP1) as a result of the increased water usage by households. Differences in flow rate were statistically significant, except between WWTP4 and WWTP1, despite WWTP4 serving a population approx. 17,500 inhabitants greater than that of WWTP1. The presence of a combined sewer network (2%) in WWTP1 may have accounted for the observed increase in wastewater flow. Chemical parameters exhibited different behaviors across WWTPs. Differences in COD and BOD_5_ concentrations were statistically significant (Kruskal–Wallis test, *p* < 0.05), showing a trend that may have been attributed to the structure of the sewer network. Higher concentrations were observed in WWTPs served by separate sewer systems, whereas the lowest values were recorded at WWTP2, whose predominantly combined network (84%) may have contributed to the dilution of sewage (WWTP4 > WWTP3 > WWTP1 > WWTP2). Conversely, NH_4_-N concentrations did not differ significantly among the WWTPs (Kruskal–Wallis test, *p* > 0.05).

The correlations among these hydrochemical parameters exhibited similar trends across the different WWTPs, as illustrated in Appendix A. In fact, regardless of the type of WWTP, chemical parameters were positively correlated with each other, whereas they showed a negative correlation with influent wastewater flow and rainfall (Figure 1). In particular, the chemical parameter that showed the highest correlation with wastewater flow was NH_4_-N, with ρ values ranging from - 0.662 in WWTP1 to - 0.753 in WWTP2.

OLS regression analysis was performed followed by calculating the DW statistic that revealed positive autocorrelation; thus, GLS regression was applied to account for both heteroscedasticity and autocorrelation (Appendix A). The analysis revealed that rain influenced flow rate to a different extent depending on the WWTP: WWTP4 exhibited the strongest relationship between rainfall and flow rate (R^2^ = 0.460), while WWTP3 had the weakest (R^2^ = 0.051) (Appendix A). Regarding chemical parameters, NH_4_-N was highly influenced by flow rate across all WWTPs (Table 4 and Appendix A), but particularly in WWTP2 (R^2^ = 0.580), which was primarily served by a combined sewer system, and WWTP4 (R^2^ = 0.492), where the flow rate was largely affected by rainfall, as previously discussed.

### 3.2. Comparison of Different SARS-CoV-2 Normalization Methods

The SARS-CoV-2 concentration data used for normalization are reported in Appendix A and depicted in Figure 2. Overall, 62.7% (160/255) of the samples were positive, with the highest detection rates observed in WWTPs located in tourist areas, namely WWTP1 (69.3%, 52/75) and WWTP4 (66.1%, 37/56), followed by WWTP3 (60.4%, 29/48) and WWTP2 (55.3%, 42/76). Dynamic normalization based on the high data approach provided viral loads in the same order of magnitude of static normalization, with median values ranging from 9.32 × 10^8^ GC/day/inhabitants (2.96 × 10^5^–2.50 × 10^9^ GC/day/inhabitants) for WWTP2 to 5.28 × 10^9^ GC/day/inhabitants (2.61 × 10^5^–1.34 × 10^10^ GC/day/inhabitants) for WWTP3. As expected, dynamic normalization based on the low data approach provided lower estimates since viral load was assumed to be equivalent to SARS-CoV-2 concentration per unit of chemical parameter, without considering other population-related factors (Figure 3).

### 3.3. Correlation Between Different SARS-CoV-2 Normalization Methods and Clinical COVID-19 Cases

The correlations between SARS-CoV-2 loads in wastewater and COVID-19 cases were evaluated on the basis of archived environmental and clinical data collected during a period when COVID-19 clinical surveillance was a reliable measure of the real disease incidence, given the public health measures in place and the high sensitivity of the methods (molecular tests) used to confirm a positive COVID-19 case (February 2021–December 2021; [26]). Given the small dataset for this timeframe, data from different WWTPs were also pooled (n = 64) (Table 5).

The results of Spearman’s analysis showed a positive correlation between SARS-CoV-2 (either non-normalized or normalized with various methods) and COVID-19 cases. In particular, the highest correlation values were obtained for SARS-CoV-2 normalization methods based on static population estimates (ρ = 0.405) and dynamic population estimates using COD or BOD_5_ (ρ = 0.378 each) when both analytical and technical data were available for viral load calculations.

Dynamic population normalization using only a low data approach resulted in a decrease in the correlation coefficient for both COD- or BOD_5_-based estimates (ρ = 0.346 and ρ = 0.356, respectively). Viral loads normalized by NH_4_-N had the lowest correlation coefficients (ρ = 0.335 using the high data approach or ρ = 0.308 using the low data approach), which was similar to the non-normalized SARS-CoV-2 values (ρ = 0.329).

## 4. Discussion

The large-scale surveillance of infectious agents can help in controlling the spread of transmissible diseases in a population. In this context, wastewater analysis represents a valuable approach compared to clinical testing, since it allows for monitoring large populations with a single analysis, considering not only symptomatic cases but also presymptomatic and asymptomatic individuals.

However, SARS-CoV-2 signals in wastewater can be influenced by various factors, particularly by the structure of the sewer network, which affects sewage composition (e.g., dilution may occur in both combined sewer systems and separate systems affected by stormwater intrusion) [24,35]. Accordingly, there is a need to normalize SARS-CoV-2 concentrations in wastewater relative to the population connected to each WWTP (i.e., those contributing to viral shedding) in order to enable meaningful comparisons of SARS-CoV-2 data across different locations and time points.

For example, the European Commission recommends European member states to calculate virus loads per capita per day by applying “normalization using 24 h wastewater flow during the time of sampling and population size of the sewershed” [21]. Alternatively, normalization approaches based on phages or PMMoV, as well as other parameters, are also recommended to account for factors causing fluctuations in the viral load.

In particular, ideal analytes for normalization should fulfill three requirements: (i) human origin, (ii) stability in sewers, and (iii) detection in the same water sample used to determine the concentration of viral RNA and, possibly, with fast and easy analytical methods [36,37]. Chemical parameters can be considered promising for normalization purposes, since they are routinely measured by the Integrated Water Service Managers as part of the regulatory monitoring of WWTPs, such as measuring water treatment efficiency (e.g., EU directive 2024/3019 concerning urban wastewater treatment). Nevertheless, such parameters are not unique to human activity or metabolism.

COD and BOD_5_ measure the amount of oxygen required for the oxidation of organic matter via chemical reactions or via microorganism-based processes, respectively. Therefore, they represent the amount of organic matter, mainly fecal matter. However, this assumption is valid for non-industrial catchment areas, where the amount of non-human substances that can influence COD values is limited, as well as the presence of compounds that might inhibit oxidizing bacteria during BOD_5_ determination [7]. NH_4_-N is a metabolite obtained from urea hydrolysis and serves as an indirect marker of urine. Thus, many authors consider this analyte to be less influenced by non-human sources compared to COD and BOD_5_ [10], although some authors have found that NH_4_-N is also influenced by industrial sources [38].

In this study, different SARS-CoV-2 normalization approaches were tested using medium-sized urban WWTPs as case studies. The strength of the relationship between environmental and clinical data were improved when viral data were normalized, consistent with the findings from other studies [10,39,40]. In particular, the highest correlation coefficient was observed for the static population estimates (ρ = 0.41; moderate correlation according to the ρ ranking by Stachler et al., 2018 [41]) and similar correlation values were also obtained when SARS-CoV-2 data were normalized by COD and BOD_5_. Thus, COD and BOD_5_ were the best chemical parameters for viral normalization to represent clinical cases, while a slightly weaker correlation was obtained using NH_4_-N-based viral estimates.

These results partially agreed with previous findings in the literature regarding SARS-CoV-2 and other respiratory viruses for which clinical surveillance is in place. For example, in the United States, Hutchison et al. (2022) [42] found similar correlation values between wastewater and clinical data, using either BOD_5_- or ammonia-normalized viral data in two urban WWTPs serving a population of approximately 100,000 inhabitants. Likewise, in Germany, Haeusser et al. (2023) [43] reported similar correlation values when SARS-CoV-2 was normalized by COD or ammonia in several medium- and small-sized WWTPs (<100,000 PE). Moreover, the COD- and BOD_5_-normalized viral loads of SARS-CoV-2, Influenza A virus, and Respiratory Syncytial Virus showed the strongest correlation with clinical case data obtained from their respective clinical surveillance systems in a large catchment area (approx. 320,000 inhabitants served by a single WWTP) located in Valencia, Spain, according to Girón-Guzmán et al. (2024) [44].

In the present study, the lower reliability of NH_4_-N as population marker can probably be attributable to the type of sewage system, as that can impact the decay rate of the analytes. In particular, rainfall events directly influenced the WWTP inlet flow rates, which were negatively correlated with chemical parameters owing to the dilution effects of sewages. However, the decay of chemical signals with an increasing flow rate was more evident for NH_4_-N compared to COD and BOD_5_. This result may be attributable to stormwater intrusion, a phenomenon that can alter the conditions within sewer pipes. For instance, it may cause slight changes in dissolved oxygen levels and facilitate the entry of ammonia-oxidizing bacteria, which are responsible for the nitrification of ammonium into nitrites and nitrates, which are compounds widely present in natural waters [45]. Nevertheless, this explanation remains mostly speculative since substance transformations within sewer pipes are rarely investigated [46]. 

Moreover, in this paper, variations in the correlation between viral loads and COVID-19 cases were observed among different WWTPs. The static normalization of viral load decreased its representativeness for clinical cases in WWTPs located in touristic areas (i.e., WWTP1 and WWTP4), likely due to fluctuations in the number of people in the catchment area. According to regional tourism statistics, the municipalities of Pisa and Viareggio, primarily contributing to WWTP1 and WWTP4, respectively, hosted approximately 720,000 and 240,000 tourist arrivals (defined as visitors staying at least one night) in 2022 [47]. Interestingly, in such WWTPs, viral normalization with chemical parameters led to an improvement in the correlation coefficient compared to the static approach. These findings were consistent with those reported by Foladori et al. (2024) [48], who observed that a chemical-based normalization approach was more effective for estimating viral circulation in tourist catchment areas, which are subject to significant population size fluctuations.

### Limitations of the Study

The results presented in this study have a site-specific validity, and their generalizability is constrained by limitations in both the environmental and clinical surveillance data.

The WBE data are representative of only a limited number of WWTPs which exhibited comparable characteristics regarding the served population (40,000–110,000 inhabitants) and influent composition, primarily comprising domestic wastewater with minimal industrial input. Thus, the obtained population normalization results are not directly transferable to WBE scenarios characterized by different sewage compositions that can affect SARS-CoV-2 signals.

Various epidemiological indicators are recommended by the WHO to assess the level of COVID-19 transmissibility, e.g., new confirmed cases, test positivity rate, hospitalization, and intensive care unit (ICU) admissions [27]. However, at the level of spatial disaggregation used in this study, only data on new confirmed cases are available. This metric, however, is a less reliable indicator of the true prevalence of infection compared to the test positivity rate, as it can be influenced by factors such as the performance of the surveillance system, testing policies, laboratory capacity, and reporting practices [27]. For this reason, correlation analyses between WBE and clinical data are conducted using a limited number of data points, specifically referring to a timeframe in which surveillance strategies and testing methods remained consistent in Tuscany (Italy). Nevertheless, given the nature of the clinical data employed, we assume no delays in reporting cases or geographical misattribution of cases, and we further assume that the testing rate remains constant throughout the selected period.

## 5. Conclusions

The surveillance of infectious diseases using clinical testing can be improved through the development of wastewater monitoring programs, where data must be carefully managed to understand the actual circulation of a pathogen across different communities and time points. Our data confirmed the role of viral wastewater data normalization in improving the correlation with clinical cases, using data from SARS-CoV-2 sewage monitoring alongside COVID-19 surveillance during a period of high testing frequency. Specifically, our results support the use of normalization based on chemical parameters, particularly COD and BOD_5_, which can be monitored in geographical areas where served population data are unavailable, flow rate measurements are unfeasible, and economic constraints limit surveillance efforts, given the affordability of such determinations.

## Figures and Tables

**Figure 1 viruses-17-00672-f001:**
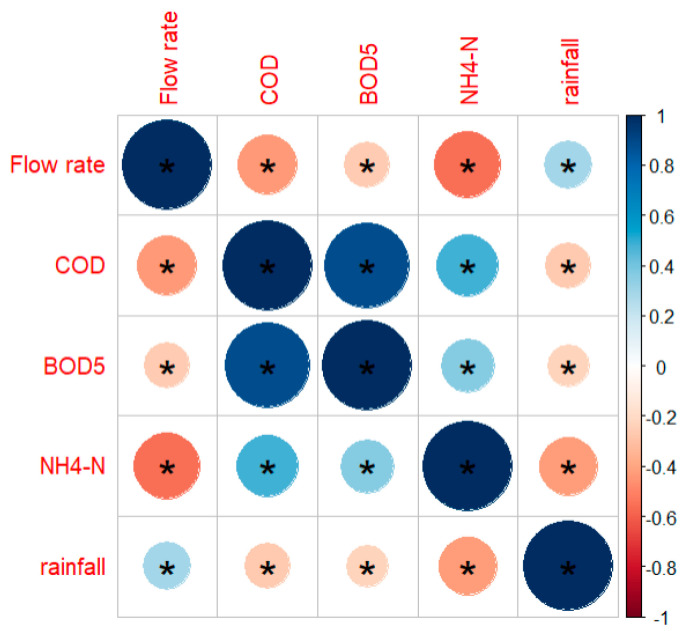
Spearman’s correlations among the chemical parameters, wastewater flow rate, and rainfall, considering all WWTP data (n = 359). Colors represent the value of Spearman’s correlation coefficient: the darker the color, the larger the correlation magnitude. An asterisk indicates statistical significance at the 0.05 level.

**Figure 2 viruses-17-00672-f002:**
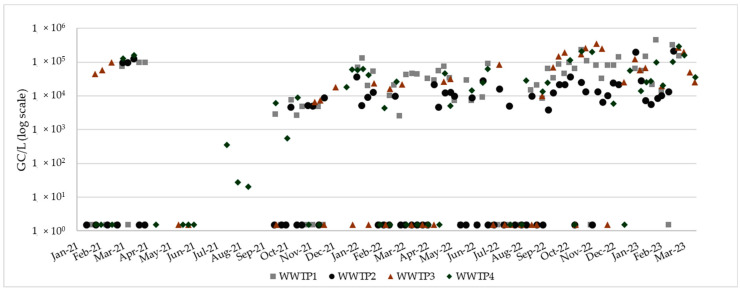
Time trend of SARS-CoV-2 concentration in the study period. WWTPs are indicated with different symbols (n = 75 WWTP1, n = 76 WWTP2, n = 48 WWTP3, n = 56 WWTP4). Samples under the limit of detection (LOD) of the analytical method were assigned a value equal to half the LOD (1.5 GC/L).

**Figure 3 viruses-17-00672-f003:**
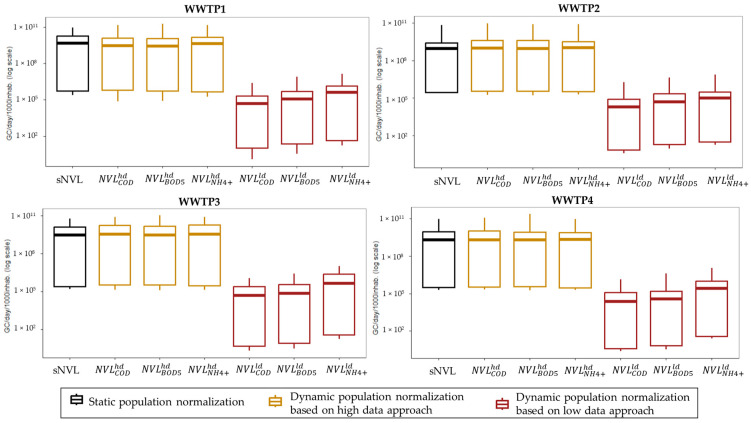
Normalized viral loads, NVL (median, first, and third quartiles, minimum and maximum) according to different normalization approaches. When SARS-CoV-2 was not detected, NVLs were calculated by assigning SARS-CoV-2 concentrations a value equal to half the detection limit. NVL data are reported separately for each WWTP, following a clockwise order: WWTP1 (n = 75), WWTP2 (n = 76), WWTP4 (n = 56), and WWTP3 (n = 48).

**Table 1 viruses-17-00672-t001:** Features of the WWTPs and sewer networks.

WWTP Type	Design Capacity (PE)	Served Population (Inhabitants)	Sewer Length (km^2^)	Served Municipalities (% of Sewage Discharged to the WWTP)
WWTP1	52,000	42,931	145	Pisa (63.2%); San Giuliano Terme (30.1%); Vecchiano (6.7%)
WWTP2	88,670	68,070	313	Empoli (59.9%); Vinci (15.5%); Montelupo Fiorentino (13.2%); Capraia e Limite (8.0%); Cerreto Guidi (2.3%); Montespertoli (1.2%)
WWTP3	120,000	110,871	387	Massa (60.4%); Carrara (31.3%); Montignoso (8.4%); Forte dei Marmi (0.03%)
WWTP4	93,000	60,262	167	Viareggio (99.7%); Massarosa (0.19%); Camaiore (0.05%); Vecchiano (0.04%)

PE = population equivalent.

**Table 2 viruses-17-00672-t002:** Site-specific average values of the daily production of the markers per person (mg/day/inhabitant) obtained during the period of national lockdown (February 2021–October 2021). The variability of the measures is expressed as the interquartile range (in brackets).

		Daily Production of Markers Per Person (mg/Day/Inhabitant)		
WWTP Type	n. obs	COD	BOD_5_	NH_4_-N
WWTP1	27	62,096 (48,297–68,772)	24,064 (18,009–29,115)	10,584 (8890–11,494)
WWTP2	29	48,476 (36,243–59,558)	18,558 (12,635–24,751)	10,392 (9792–11,979)
WWTP3	11	68,695 (52,040–75,446)	41,014 (33,359–51,303)	7700 (5295–9232)
WWTP4	18	84,698 (75,148–95,138)	55,204 (48,241–65,991)	8123 (6943–9288)

**Table 3 viruses-17-00672-t003:** Chemical parameters and wastewater flow rates of the WWTPs. Total number of observations for each WWTP (n) is reported. Data are summarized as median and interquartile range.

WWTP Type	n	Wastewater Flow (L/Day)	COD (mg/L)	BOD_5_ (mg/L)	NH_4_-N (mg/L)
WWTP1	91	1.02 × 10^7^ (9.05 × 10^6^–1.25 × 10^7^)	281 (234–362)	120 (88–155)	50 (40–56)
WWTP2	93	1.41 × 10^7^ (1.20 × 10^7^–1.84 × 10^7^)	195 (141–259)	75 (50–99)	46 (37–51)
WWTP3	85	1.84 × 10^7^ (1.62 × 10^7^–2.15 × 10^7^)	373 (286–450)	265 (180–340)	46 (35–55)
WWTP4	90	1.11 × 10^7^ (1.02 × 10^7^–1.25 × 10^7^)	505 (428–590)	320 (260–380)	47 (41–53)

**Table 4 viruses-17-00672-t004:** Determination coefficient (R^2^) from GLS models relating WWTP influent flow rates to chemical parameters (COD, BOD_5_, and NH_4_-N).

WWTP Type	n	COD	BOD_5_	NH_4_-N
WWTP1	91	0.1776	0.0721	0.3445
WWTP2	93	0.2838	0.1647	0.5795
WWTP3	85	0.2953	0.3824	0.3999
WWTP4	90	0.2417	0.2562	0.4915
Pooled WWTPs	359	0.1776	0.0721	0.3445

**Table 5 viruses-17-00672-t005:** Spearman’s correlation coefficients (ρ) for clinical COVID-19 cases against non-normalized SARS-CoV-2 values and normalized SARS-CoV-2 loads with static or dynamic approaches (data refer to a period when COVID-19 clinical surveillance was considered reliable of the real disease incidence in Italy; February 2021–December 2021). Spearman’s correlation coefficients with *p*-values < 0.1 are in bold.

Normalization Approaches	WWTP1 (n = 18)	WWTP2 (n = 19)	WWTP3 (n = 10)	WWTP4 (n = 17)	Pooled WWTPs (n = 64)
SARS-CoV-2 concentration (no normalization)	0.169	0.335	**0.828**	0.120	**0.329**
Static normalization	0.073	**0.604**	**0.851**	0.201	**0.405**
Dynamic COD normalization, low data approach ($NVL_{COD}^{ld}$)	0.191	**0.391**	**0.875**	0.287	**0.346**
Dynamic BOD_5_ normalization, low data approach ($NVL_{BOD5}^{ld}$)	0.112	0.275	**0.888**	0.287	**0.356**
Dynamic NH_4_+ normalization, low data approach ($NVL_{NH4+}^{ld}$)	0.166	0.321	**0.802**	0.120	**0.308**
Dynamic COD normalization, high data approach ($NVL_{COD}^{hd}$)	0.191	**0.391**	**0.875**	0.287	**0.378**
Dynamic BOD_5_ normalization, high data approach ($NVL_{BOD5}^{hd}$)	0.112	0.275	**0.888**	0.287	**0.378**
Dynamic NH_4_+ normalization, high data approach ($NVL_{NH4+}^{hd}$)	0.166	0.321	**0.802**	0.120	**0.335**

## Data Availability

The original contributions presented in this study are included in the article/Appendix A. Further inquiries can be directed to the corresponding author.

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
