# Peer review of "Evaluating Population Normalization Methods Using Chemical Data for Wastewater-Based Epidemiology: Insights from a Site-Specific Case Study"

_viruses, 2025, doi:10.3390/v17050672_

Round 1
Reviewer 1 Report
Comments and Suggestions for Authors
Please see the word file attached.

Comments on the Quality of English Language
Since I am not a native speaker myself, I cannot review the English sufficiently competently.
Author Response
Please consider the attached point-by-point rebuttal letter

Reviewer 2 Report
Comments and Suggestions for Authors
The authors have conducted a study assessing the potential for dynamic or statis population estimates to inform wastewater-based epidemiology. I found the article to be relevant and timely – the question of normalizing the measurement of a pathogen in wastewater to some other measure is an important area of study and also one that is not quite settled. The authors have done well to describe their study and I have only a few minor suggestions and questions:
- In the methods it was not clear how frequently the wastewater was sampled. The authors report the study was conducted from February 2021 – March 2023, but how frequently was the wastewater taken from the different wastewater treatment plants? It would be nice to have a figure showing the trends in the amount of SARS-coV-2 RNA isolated from the wastewater over the course of the study. That would give an indication of whether the samples were continuously taken during the time period or pulled during more concentrated time periods.
- In the introduction the authors do well to identify how wastewater may not relate to clinical case data, due to people moving about and what is measured. The authors could also mention the uncertainty around clinical data – that cases may be misattributed to home addresses, that the time of test result may not represent the time of peak shedding, etc.
- The linear regression analyses that the authors conducted likely have serial autocorrelation. Have they done a Durbin-Watson test to see, and then perhaps switched to an OLS method to account for the serial autocorrelation?
- Figure 1 shows significant heaping at zero. Would the authors consider adjusting that analysis to look at the relationship between rain and these parameters only if there were rain? That is to say what does the relationship look like and what is the correlation if you remove the sampling days with no rain?
- Figure 2 suggests the relationships are pretty similar across the four treatment plants. Would the authors consider condensing that to a single figure (easier for the reader) and then including the treatment-plant specific figures in the supplementary documents?
- Lines 308-311 don’t quite make sense to me. If the transmission is similar between communities, the normalized levels of SARS-CoV-2 should be similar also.
Author Response

(The authors gave the same response as above.)

Reviewer 3 Report
Comments and Suggestions for Authors
GENERAL COMMENTS
The authors have addressed the issue of methods for normalizing the results obtained with WBE. This problem has been particularly discussed following the development of WBE during the Covid-19 pandemic.
Although limited to four WWTPs, the study presents interesting results as it compares the static method with the dynamic one, based on the use of common physico-chemical parameters that are routinely analyzed in wastewater treatment influent. What is particularly interesting is that this assessment is conducted in plants with different characteristics.
The authors then evaluate the correlation between normalized viral loads and clinical Covid-19 cases using the two normalization approaches and conclude that COD and BOD5 offer a viable alternative for viral normalization in WBE studies when population estimates or flow rate measurements are unavailable.
SPECIFIC COMMENTS
p 3 line 97 It is suggested to better describe what is meant by unit of biomarker or to provide an example.
Add to the caption of Table 1 an explanation of what the percentage value refers to.
P 5 lines 182-184 “Cc,i is the concentration of each chemical parameter, and 1000 is a constant used to represent the viral load of 1000 inhabitants. Authors should add an example of how the parameter concentration is expressed, as is done for the concentration of SARS-CoV-2 (GC/L).
P6, lines 216-219: Why do the authors report “to estimate the number of new positive cases within each WWTP’s catchment area” instead of referring to the total number of cases to correlate with the viral load obtained through WBE?
In Figure 2, the legend with colours and correlation values is not very clear. It is suggested to use a single, larger pattern placed at the centre of the figure.
P9, line 285: Is Table 5 in the supplementary materials?
P11, lines 343-346: “Thus, COD and BOD5 were the best chemical parameters for viral normalization to represent clinical cases, while a slightly weaker correlation was obtained using NH4-N-based viral estimates.” Is this finding specific to SARS-CoV-2, or are there similar reports in the literature for other viruses?
Author Response

(The authors gave the same response as above.)

Round 2
Reviewer 1 Report
Comments and Suggestions for Authors
No further comments.